# AQMDRL: Automatic Quality of Service Architecture Based on Multistep Deep Reinforcement Learning in Software-Defined Networking

**DOI:** 10.3390/s23010429

**Published:** 2022-12-30

**Authors:** Junyan Chen, Cenhuishan Liao, Yong Wang, Lei Jin, Xiaoye Lu, Xiaolan Xie, Rui Yao

**Affiliations:** 1School of Computer Science and Information Security, Guilin University of Electronic Technology, Guilin 541004, China; 2School of Information and Communication, Guilin University of Electronic Technology, Guilin 541004, China; 3School of Computer Science and Engineering, Northeastern University, Shenyang 110819, China

**Keywords:** software-defined networking, deep reinforcement learning, quality of service, deep deterministic policy gradient, multistep, SumTree

## Abstract

Software-defined networking (SDN) has become one of the critical technologies for data center networks, as it can improve network performance from a global perspective using artificial intelligence algorithms. Due to the strong decision-making and generalization ability, deep reinforcement learning (DRL) has been used in SDN intelligent routing and scheduling mechanisms. However, traditional deep reinforcement learning algorithms present the problems of slow convergence rate and instability, resulting in poor network quality of service (QoS) for an extended period before convergence. Aiming at the above problems, we propose an automatic QoS architecture based on multistep DRL (AQMDRL) to optimize the QoS performance of SDN. AQMDRL uses a multistep approach to solve the overestimation and underestimation problems of the deep deterministic policy gradient (DDPG) algorithm. The multistep approach uses the maximum value of the n-step action currently estimated by the neural network instead of the one-step Q-value function, as it reduces the possibility of positive error generated by the Q-value function and can effectively improve convergence stability. In addition, we adapt a prioritized experience sampling based on SumTree binary trees to improve the convergence rate of the multistep DDPG algorithm. Our experiments show that the AQMDRL we proposed significantly improves the convergence performance and effectively reduces the network transmission delay of SDN over existing DRL algorithms.

## 1. Introduction

In the era of the Internet of Things (IoT), sensors can connect any physical device to the Internet, creating a digital and intelligent world. Individuals, businesses and industries can work around the clock with vast amounts of data collected and implemented by IoT sensors, leading to the exponential growth of Internet traffic. Thus, owing to the unprecedented growth of Internet traffic and the stringent quality of service (QoS) requirements of various applications, traditional data center networks are under tremendous pressure.

Software-defined networking (SDN) has become one of the critical technologies for data center networks, with features such as decoupling data plane and control plane, controller management of global network and programmability, etc. SDN controllers can improve the network performance from a global perspective using artificial intelligence algorithms, including network throughput, packet loss, transmission delay, and load balancing, which is of great significance to ensure the network QoS [1]. However, most traditional QoS optimization schemes in SDN are based on traditional routing or heuristic algorithms [2,3,4,5,6]. Heuristic algorithms need to build models based on specific real-world problems. When faced with dynamically changing complex networks, heuristic algorithms are often powerless to guarantee network QoS.

In recent years, many researchers have applied deep learning (DL) to the QoS optimization of SDN due to its powerful learning capability and excellent performance advantages [7,8,9,10,11]. However, deep learning requires large datasets for model training and has poor generalization ability, making dynamic network performance optimization challenging. Compared to deep learning, reinforcement learning (RL) employs online learning to train a model by continuously exploring, learning and changing its behavior to obtain the best return. Reinforcement learning can generate action decisions based on the state of the environment and reward feedback, and has strong decision-making and generalization capabilities. Therefore, some researchers have applied reinforcement learning to the QoS optimization of networks, allowing the agent to interact with the network environment, in order to achieve the adaptive optimization of network performance [12,13,14,15,16,17,18]. However, the perceptual capability of RL algorithms is weak, which makes the decisions of RL algorithms in dynamic network scenarios highly variable and extends the learning time. Deep reinforcement learning (DRL) combines the perceptual capability of DL and can effectively improve the convergence performance of RL algorithms. Thus, it has attracted the attention of many researchers and has been applied to SDN performance optimization, such as deep Q-network (DQN) [19,20,21,22], deep deterministic policy gradients (DDPG) [23,24,25,26,27,28,29], twin delayed deep deterministic policy gradient (TD3) [30,31], etc. However, the DQN algorithm can only make decisions in a discrete action space and has limited capability. The slow and unstable convergence of DDPG and TD3 algorithms will lead to more extended trial-and-error times, thus making the network jittery for a long time. It leads to poor QoS performance and seriously affects the regular order of the network.

Concerning the above challenges, we propose an automatic QoS architecture based on multistep DRL (AQMDRL) to optimize the QoS performance of SDN in load balancing and network latency. The agent of AQMDRL first receives the network link status and the evaluated reward value. We use the link delay and the standard deviation of link bandwidth as the agent’s reward function. Then, it generates an action, namely the interconnection weight matrix. Finally, it uses the Dijkstra algorithm to select the optimal routing path through the interconnection weight matrix and then creates the forwarding flow table. The agent evaluates the quality of the action and determines the next action through the reward value in each episode to minimize the link delay and achieve link load balance. To avoid long-term network jitter before convergence, the AQMDRL algorithm aims to solve the problem of slow convergence rate and instability of traditional DRL algorithms applied in QoS optimization of SDN. In order to make the algorithm more stable after convergence, it uses a multistep strategy to optimize the DDPG algorithm, which effectively alleviates the overestimation and underestimation problem of DDPG. In addition, it further adopts the SumTree-based prioritized experience sampling mechanism to implement the experience sampling of DDPG to improve the convergence rate. Through experiments, it is verified that the proposed optimization scheme can effectively improve the convergence rate and stability, and reduce the impact of network performance caused by the reinforcement learning model in the trial-and-error phase.

The main contributions of this paper are as follows:We propose an automatic QoS architecture based on AQMDRL to effectively solve the poor service quality caused by unstable SDN performance.We design a DRL-based QoS optimization model to evaluate the SDN network quality of service.We use a multistep strategy to optimize the DDPG algorithm. The multistep approach uses the maximum value of the n-step action currently estimated by the neural network instead of the one-step Q-value function, as it reduces the possibility of positive error generated by the Q-value function and can effectively improve convergence stability. In addition, we adapt a prioritized experience sampling based on SumTree binary trees to improve the convergence rate of the multistep DDPG algorithm.We verify the effectiveness of the AQMDRL algorithm by designing an experimental system.

The rest of this paper is organized as follows: Section 2 entails the related research; Section 3 describes the network architecture and the AQMDRL algorithm; Section 4 discusses the experiments and performs the analysis; Section 5 concludes the paper.

## 2. Related Research

Traditional network-based QoS policies are usually based on open shortest path first (OSPF) [2], equal-cost multipath routing (ECMP) [3] or heuristic optimization algorithms [4,5,6] for forwarding traffic. However, OSPF and ECMP use fixed forwarding methods, which easily result in link congestion when faced with complex traffic tasks. Heuristic algorithms need to build models based on specific real-world problems. When faced with dynamically changing complex networks, heuristic algorithms are often powerless and have limited scalability to guarantee network service quality.

Traditional network-based QoS policies cannot handle increasingly complex tasks and no longer meet the needs of SDN environments. In recent years, DL has been increasingly applied to SDN networks due to its powerful learning algorithms and excellent performance advantages. Zou et al. [7] proposed a deep learning approach to perform time-aware service QoS prediction tasks through feature integration. In other studies [8,9,10], a long short-term memory neural network algorithm (LSTM) was used to optimize the SDN network performance. Chen et al. [11] employed the LSTM traffic prediction method in the SDN application plane to solve the SDN load balancing problem. Nonetheless, deep learning requires a large number of datasets for training, and it has poor generalization ability, arising from its inability to interact with the environment. The above reasons make it difficult to optimize the performance of dynamic networks. Compared with DL, RL uses online learning for model training, changing the agent behavior through continuous exploration and experimentation to obtain the best reward; therefore, it does not need to train the model in advance, but can change its action according to the environment state and reward feedback. Reinforcement learning has strong decision-making and generalization ability. Certain authors [12,13,14] have combined RL with SDN to optimize QoS. Younus et al. [15] used the Q-learning algorithm to optimize the network performance and effectively improve the network convergence speed. They [16] added QoS to the reward function setting. Casas-Velasco et al. [17] proposed the Reinforcement Learning and Software-Defined Networking Intelligent Routing (RSIR) algorithm, which avoids links with high latency and packet loss by minimizing the reward value of the Q-learning agent. The RSIR algorithm outperforms Dijkstra’s algorithm regarding link throughput, packet loss and latency. Al-Jawad et al. [18] implemented a trade-off between QoS and quality of experience (QoE) for users of the Q-learning algorithm. However, the Q-learning algorithm learns the optimal action-value function as a Q-table, and does not access previous experience, resulting in a lack of data representation in deep learning.

Deep reinforcement learning utilizes the data representation capabilities of deep neural networks in reinforcement learning to implement decisions, which opens up a new avenue to solve complex network QoS problems. Baek et al. [19] used a DQN-based optimization method to approximate the optimal value function, in order to maximize and stably converge the common objective of the nodes. Chen et al. [20] combined a graph neural network and ensemble learning with the DQN algorithm to address the lack of generalization ability in untrained SDN topologies. Other scholars [21,22] combined DQN and LSTM to achieve network routing policy optimization. However, the action space of DQN is discrete, so they cannot be solved for a high-dimensional set of states. DDPG solves the problem that DQN can only make decisions in discrete action spaces. In further studies [23,24,25], DDPG was applied to SDN routing optimization, and the scheme achieved intelligent optimization of the network and effectively reduced network latency. Lan et al. [26] proposed the QoS optimization algorithm, R-DRL, which combines the DDPG algorithm with LSTM to generate a dynamic traffic scheduling policy that satisfies the objective of QoS optimization. Guo et al. [27] proposed a DDPG-based DQSP protocol that guarantees security and QoS in the network. Mai et al. [28] proposed a DDPG-based slicing method to provide different QoS guarantees. Chen et al. [29] updated the sampling method of the experience pool using the SumTree structure, which improves the random sampling strategy of the experience replay mechanism in DDPG. Although the DDPG algorithm can learn more efficiently on subsequent actions, it blindly selects the action with the largest Q-value when choosing the action, causing the DDPG to suffer from an overestimation problem.

In order to solve the overestimation problem of the DDPG algorithm, Fujimoto et al. [30] proposed the TD3 algorithm, which refers to the clipped double Q-learning algorithm in the value network and uses delayed policy update and target policy smoothing techniques. TD3 improves the high estimation and high variance present in DDPG. Sun et al. [31] proposed a TD3-based intelligent routing technique that can self-adapt to dynamic network environments. Other scholars [32,33,34] used pinning control to address the poor scalability and low robustness problems that commonly exist in the current DRL algorithm applied to network routing policy generation.

Although the TD3 algorithm alleviates the overestimation problem, it may lead to significant underestimation bias and affect the convergence performance when using the minimum approach for value interception. TD3 still suffers from slow convergence and instability, seriously affecting the network QoS. Pan et al. [35] used the Boltzmann softmax operator to estimate the value function to solve the problem of underestimation of TD3. However, this method increases the complexity of the TD3 algorithm and requires more computing resources. Huang et al. in [36] used a Graph Convolutional Network (GCN) to optimize DRL networks. Wang et al. [37] proposed a reinforcement learning-based relay Graph Neural Network (GNN). The model combining GNN and RL performs better in specific network scenarios. However, it must simultaneously optimize both networks’ parameters, requiring a longer training time. Meng et al. [38] proposed a mixed multistep DDPG approach (MMDDPG) and empirically demonstrated that the multistep approach can effectively mitigate the overestimation problem in DRL. Inspired by this strategy, we propose an improving DDPG algorithm using multi-step method and prior experience sampling to optimize the QoS performance of SDN. It enhances the algorithm’s stability by estimating the target Q-value update network after n steps and improves the algorithm’s convergence rate by extracting samples with a more influential temporal-difference error (TD error), thus solving the problems of unstable network performance and poor quality of service.

## 3. System Architecture and Components

The DRL-based QoS optimization architecture includes a data plane, a control plane and an application plane, as shown in Figure 1. The specific functions of each plane are as follows:

Data plane. The data plane consists of SDN-enabled switches responsible for network data transmission. It senses the network state in real time and passes the network state to the control plane through the southbound interface. Meanwhile, it receives policies issued by the control plane. When a new flow arrives, the switch queries the flow table entries according to priority and matches the data one by one. Then, it performs the corresponding processing step according to the matching rules and results. If the flow does not correspond to any flow table entry, it will be discarded or reported to the controller.Control plane. The control plane consists of the SDN controllers responsible for flow table configuration. It passes the network state collected from switches to the application plane. Next, it receives the interlink weight matrix generated by the application plane. Finally, it uses the Dijkstra algorithm to select the optimal routing path and sends the forwarding flow table to the switches.Application plane. The application plane runs the AQMDRL agent responsible for generating routing optimization policies. The AQMDRL agent uses the network state as the state input and communicates the generated interlink weight matrix to the control plane. This process is repeated until the agent learns the action strategy that maximizes the reward value.

## 4. Problem Definition and System Model

Based on the system architecture described above, the role of the DRL agent is to assign routes for incoming traffic demands to maximize network utility. In this section, we describe the DRL-based QoS optimization model of the network performance evaluation.

We use G to refer to a network in the data plane, as shown in Equation (1):(1)G=(V,E),
where V represents the set of forwarding nodes, as shown in Equation (2); E represents the set of links, as shown in Equation (3).
(2)$$V=\left[v_{1},\;v_{2},\ldots ,\;v_{n}\right],$$
(3)$$E=\left[e_{1},e_{2},\ldots ,e_{m}\right]$$

We use $c_{i}$ to denote the capacity of link $e_{i}$ and define *C* as the set of network link capacities, as shown in Equation (4), where |C|=|E|=m.
(4)$$C=\left[c_{1},c_{2},\ldots ,c_{m}\right].$$

We use $b_{ij}^{t}$ to denote the traffic demand from node $v_{i}$ to node $v_{j}$ at time t and define $B_{t}$ as the set of all traffic demands, as shown in Equation (5):(5)$$B_{t}=\left[\begin{array}{ccc} b_{11}^{t} & \cdots & b_{n1}^{t}\\ \vdots & \ddots & \vdots \\ b_{1n}^{t} & \cdots & b_{nn}^{t} \end{array}\right].$$

We define $\varphi_{t}$ as the link-delay coefficient at time t, which is calculated by Equation (6):(6)$$\varphi_{t}=\sum_{i=1}^{m}\frac{1}{d_{i}^{t}},$$
where $d_{i}^{t}$ represents the delay of link $e_{i}$ at time t. The higher the link-delay coefficient, the lower the transmission delay of the link, and the faster the transmission.

We define $\rho_{t}$ as the load-balancing coefficient at time t, which is calculated by Equation (7):(7)$$\rho_{t}=\sum_{i=1}^{m}\sigma \left(u_{i}^{t}\right),$$
where $u_{i}^{t}$ represents the bandwidth usage of link $e_{i}$ at time t, and σ(·) represents the standard deviation. The load-balancing coefficient is used to examine the load-balancing degree of the network. The smaller the load-balancing coefficient, the better the load-balancing degree.

Reinforcement learning is a learning process for the agent to achieve an objective in interacting with the environment. The agent generates an action for the environment. Then, the environment evaluates the agent according to the goal, stimulating the agent to generate the next action. In this paper, the optimization objective is to maximize the link-delay coefficient $\varphi_{t}$ and minimize the load-balancing coefficient $\rho_{t}$, which can optimize the QoS performance of SDN in load balancing and network latency.

Given the above optimization objective, we design the state, action, and reward of reinforcement learning as follows:
1.State. We set the state as the network state information obtained from the environment. In this paper, the state corresponds to the traffic demand $B_{t}$, as shown in Equation (8).
(8)$$s_{t}=\left[B_{t}\right]$$
2.Action. We design the action as the set of link weights generated by the DRL agent at time t, as shown in Equation (9).

(9)$$a_{t}=\left[w_{1},w_{2},\ldots ,w_{m}\right]$$
where $w_{i}$ denotes the weight value of link $e_{i}$.


3.Reward. The reward function is the return for the previous action and is used to evaluate the service quality. According to the optimization objective, we design the reward function as Equation (10), where $\beta_{1}$ and $\beta_{2}$ denote the weight coefficients of the reward. To make sure that the value of the link-delay coefficient $\varphi_{t}$ and the load-balancing coefficient $\rho_{t}$ are of the same order of magnitude, we use $\log_{2}\left(\cdot \right)$ to process the link-delay coefficient.
(10)$$r_{t}=\beta_{1}{*\log}_{2}\left(\varphi_{t}\right)-\beta_{2}*\rho_{t}.$$


## 5. AQMDRL Algorithm

In this section, we describe in detail the AQMDRL algorithm proposed in this paper and the working process of the AQMDRL agent.

### 5.1. Deep Deterministic Policy Gradients

DDPG is an off-policy deep reinforcement learning algorithm. It is essentially the actor-critic-based framework, which combines the deterministic policy gradient and DQN based on the action value. It constructs a deterministic strategy to maximize the Q-value by using the method of gradient rise.

The DDPG algorithm contains an experience pool and four neural networks, as depicted in Figure 1. The functions are described in detail as follows:

1.Experience pool. The experience pool is used to store the tuple $\left(s_{t},a_{t},r_{t},s_{t+1}\right)$ generated during training, and the target networks randomly select mini-batch samples from the experience pool to train the model.

2.Actor online network. This network is responsible for generating action $a_{t}$ according to state $s_{t}$, noise N and policy function $\mu_{\theta }\left(\cdot \right)$, as shown in Equation (11), and interacting with the environment to obtain new state $s_{t+1}$ and reward $r_{t}$.

(11)$$a_{t}=\mu_{\theta }\left(s_{t}\right)+N$$
where θ represents the actor online network parameter.

3.Actor target network. This network is responsible for generating optimal action ${\hat{a}}_{t}$ for the critic target network according to policy function $\mu_{\theta^{\prime }}\left(\cdot \right)$ and state $s_{t}$ sampled from the experience pool, as shown in Equation (12), where $\theta^{\prime }$ represents the actor target network parameter. The parameter $\theta^{\prime }$ is periodically copied from θ.
(12)$${\hat{a}}_{t}=\mu_{\theta^{\prime }}\left(s_{t}\right)$$

4.Critic online network. This network generates the online Q-value $Q_{\omega }\left(s_{t},a_{t}\right)$ according to the online Q-value function $Q_{\omega }\left(\cdot \right)$, where ω represents the critic online network parameter. The policy gradient function uses the online Q-value to update actor online network parameter θ, as shown in Equation (13).

(13)$$\nabla_{\theta }J=\frac{1}{N}\sum_{i=0}^{N-1}\nabla_{a}Q_{\theta }\left(s,a\right){|}_{s=s_{i},a=\mu \left(s_{i}\right)}\nabla_{\theta }\mu_{\theta }\left(s\right){|}_{s_{i}}.$$
where N represents the number of mini-batch samples.

5.Critic target network. This network calculates the target Q-value $y_{i}$ according to the target Q-value function $Q_{\omega^{\prime }}\left(\cdot \right)$, as shown in Equation (14):

(14)$$y_{i}=r_{t}+\gamma Q_{\omega^{\prime }}\left(s_{t+1},{\hat{a}}_{t+1}\right).$$
where $\omega^{\prime }$ represents the critic target network parameter and γ is the reward-discount factor, γ∈[0, 1]. State $s_{t}$ and reward $r_{t}$ are obtained in the experience pool. The network parameter $\omega^{\prime }$ is periodically copied from ω. The loss function uses the target Q-value to update critic online network parameter ω, as shown in Equation (15):(15)$$L\left(\omega \right)=\frac{1}{N}\sum_{i=0}^{N-1}{\left(y_{i}-Q_{\omega }\left(s_{i},a_{i}\right)\right)}^{2}.$$
where N represents the number of mini-batch samples.

The parameters $\theta^{\prime },\omega^{\prime }$ in the target network use the soft update method for updating, as shown in Equations (16) and (17):(16)$$\theta^{\prime }=\tau \theta +\left(1-\tau \right)\theta^{\prime }.$$
(17)$$\omega^{\prime }=\tau \omega +\left(1-\tau \right)\omega^{\prime }.$$
where the update rate τ is a hyperparameter.

### 5.2. Multistep DDPG

Although the DDPG algorithm can learn more efficiently on subsequent actions, it blindly selects the action with the largest Q-value when choosing the action, causing the DDPG to suffer from an overestimation problem. Because the critical network of DDPG employs the DQN algorithm, each time, the DDPG algorithm obtains the target Q-value $y_{i}$ according to the current reward $r_{i}$ and the target Q-value $Q_{\omega^{\prime }}\left(s_{t+1},{\hat{a}}_{t+1}\right)$. However, $Q_{\omega^{\prime }}\left(s_{t+1},{\hat{a}}_{t+1}\right)$ itself calculated by the critical target network may generate forward or reverse errors. In the updating mode of the DQN algorithm, the neural network will accumulate positive errors; therefore, when the DDPG algorithm updates the parameters, $Q_{\omega^{\prime }}\left(s_{t+1},{\hat{a}}_{t+1}\right)$ may be overestimated. Similarly, if $Q_{\omega^{\prime }}\left(s_{t+1},{\hat{a}}_{t+1}\right)$ is used as the update target to update the online Q-value $Q_{\omega }\left(s_{t},a_{t}\right)$ in the previous step, $Q_{\omega }\left(s_{t},a_{t}\right)$ will also be overestimated, and such errors will gradually accumulate. The larger the action space, the more serious the overestimation problem of the DDPG algorithm will be. The inaccurate estimation of the target Q-value may make the learning strategy strongly deviate from the optimal direction, resulting in poor convergence performance.

The TD3 algorithm proposed in [30] uses the clipped double Q-learning algorithm, delayed policy update and target policy smoothing in the value network to solve the overestimation problem of the DDPG algorithm. However, it may lead to significant underestimation bias and affect the convergence performance when using the minimum approach for value interception.

This paper uses the multistep DDPG (MDDPG) method to alleviate the overestimation and underestimation of the target Q-value function of DDPG. In the MDDPG, the corresponding optimality equation for the target Q-value can be defined as the sum of the n-step discounted accumulated reward and target Q-value function $Q_{\omega^{\prime }}\left(\cdot \right)$ through the n-step forward operation, as shown in Equation (18), where n represents the number of steps, and M is the size of the experience pool.
(18)$$y_{i}^{\left(n\right)}=\{\begin{array}{c} \overset{n}{\underset{k=0}{\sum }}\gamma^{k}r_{i+k}+\gamma^{n}\underset{\hat{a}}{\max}Q_{\omega^{\prime }}\left(s_{i+n},\hat{a}\right)\;\;\;if\;i+n\leq M\\ \overset{n+i-N}{\underset{k=0}{\sum }}\gamma^{k}r_{i+k}\;\;\;\;if\;i+n>M \end{array}.$$

In Equation (18), the MDDPG algorithm continuously samples the n tuples $\left\{\left(s_{i},\;a_{i},r_{i},s_{i+1}\right),\left(s_{i+1},\;a_{i+1},r_{i+1},s_{i+2}\right),\ldots ,\left(s_{i+n-1},\;a_{i+n-1},r_{i+n-1},s_{i+n}\right)\right\}$ in the experience pool to calculate the target Q-value, thus improving the accuracy of Q-value prediction. In the previous analysis, the main reason for the overestimation of DDPG is the Q-value function $Q_{\omega^{\prime }}\left(\cdot \right)$. In MDDPG, we use the maximum value of the n-step action currently estimated by the neural network instead of the Q-value function $Q_{\omega^{\prime }}\left(\cdot \right)$, as it reduces the possibility of positive error generated by $Q_{\omega^{\prime }}\left(\cdot \right)$. In addition, we use the n-step discounted accumulated reward instead of current reward $r_{i}$ to alleviate the undervaluation problem.

According to the above target Q-value in Equation (18), we update the loss function of DDPG (Equation (15)), as shown in Equation (19):(19)$$L\left(\omega \right)=\frac{1}{N}\sum_{i=0}^{N-1}{\left(y_{i}^{\left(n\right)}-Q_{\omega }\left(s_{i},a_{i}\right)\right)}^{2}.$$

### 5.3. Priority Experience Playback Mechanism Based on SumTree

Convergence rate is one of the essential indicators of reinforcement learning performance. The DDPG algorithm implements the experience playback mechanism through the random sampling method, so the sampling probability of each sample in the experience pool will be the same. This leads to the low utilization rate of valuable samples, thus reducing the convergence rate of the algorithm.

In order to improve the convergence rate of the MDDPG algorithm, we use a prioritized mechanism based on the SumTree structure to optimize the experience pool sampling method of MDDPG. The SumTree structure contains leaf nodes, parent nodes and a root node. We calculate the corresponding priority pi for each tuple in the experience pool and stores it in the leaf node. The parent nodes store the sum of the stored values of the two child nodes connected to it, while the root node holds the sum of all leaf node storage values. The RL agent uniformly takes the value in the interval of the root node storage value and samples the experience pool data through the SumTree data extraction method, as shown in Algorithm 1. In the SumTree architecture, the higher the priority pi, the greater the probability of being sampled, thus improving the utilization of significant samples in the experience pool.
**Algorithm 1**: SumTree-based data extraction algorithm1: Uniform sampling of data at the root node, assuming as *p*2: Take the root node as the parent node and traverse its child nodes3: **repeat**4:    Define pL as the value stored on left child node;5:    **if**
pL>p **then**6:       Take the left child node as the parent node and traverse its child nodes;7:    **else**8:       p=p−pL;9:       Select the right child node as the parent node and traverse its child nodes;10:    **end if**11: **until** the selected child node is leaf node;12: Extract the tuple corresponding to the selected leaf node from experience pool.

In the DDPG algorithm, the TD error, which is the difference between the current Q-value and its next bootstrap estimate, is a critical indicator for judging the importance of samples in the experience pool. The more significant the TD error of a sample, the lower the utilization of this sample and the greater the potential to improve the performance of the neural network. Therefore, we select the TD error as the sample priority value $p_{i}$, as shown in Equation (20), where $y_{i}^{\left(n\right)}$ represents the n-step target Q-value.
(20)$$p_{i}=\left|y_{i}^{\left(n\right)}-Q_{\omega }\left(s_{i},a_{i}\right)\right|.$$

The sampling probabilities of each sample P(i) and importance sampling weights $w_{i}$ are shown in Equations (21) and (22), respectively:(21)$$P\left(i\right)=\frac{p_{i}}{\sum_{j=1}^{N}p_{j}}.$$
(22)$$w_{i}={\left(N*P\left(i\right)\right)}^{-1}/\underset{j}{\max}\left(w_{j}\right).$$
where N is the number of samples. Next, we add importance sampling weights to update the policy gradient function (Equation (13)) and loss function (Equation (19)), as shown in Equations (23) and (24), respectively:(23)$$\nabla_{\theta^{\mu }}J=\frac{1}{N}\sum_{i=0}^{N-1}w_{i}\left(\nabla_{a}Q_{\theta }\left(s,a\right){|}_{s=s_{i},a=\mu \left(s_{i}\right)}\nabla_{\theta }\mu_{\theta }\left(s\right){|}_{s_{i}}\right).$$
(24)$$L\left(\omega \right)=\frac{1}{N}\sum_{i=0}^{N-1}w_{i}{\left(y_{i}^{\left(n\right)}-Q_{\omega }\left(s_{i},a_{i}\right)\right)}^{2}.$$

### 5.4. Working Process of the AQMDRL Agent

The working process of the AQMDRL agent at each iteration is described in Algorithm 2 below.
**Algorithm 2**: AQMDRL training algorithm**Input:**Reward discount factor γ, target update rate τ, number of mini-batch samples N, number of episodes M.Initialize the parameters θ and ω;Copy the parameters θ′←θ and ω′←ω;Initialize replay buffer and set the priority of all leaf nodes in SumTree to 0;Initialize the action to explore random noise N;Initialize the state s1 collected from the SDN controller;1: **for** t = 1 to M **do**2:     Select the action weight at=μ(st)+N according to the state st in the actor online network;3:     Deploy at on the SDN controller;4:     Recalculate the forwarding paths of all traffic demands and issue the flow table through the SDN controller;5:     Obtain reward rt and new state st+1 from the SDN controller;6:     Store the tuple (st, at, rt, st+1,) into the replay buffer;7:     Calculate the sample priority value pi;8:     Update all nodes’ value of SumTree;9:     Extract n-step samples from the replay buffer through Algorithm 1;10:    Calculate the importance-sampling weight using Equation (22);11:    Calculate the target Q-value using Equation (18);12:    Use loss function (Equation (24)) to update the parameter ω;13:    Use policy gradient function (Equation (23)) to update the parameter θ;14:    Update the target network parameters:θ′←τθ+(1−τ)θ′ω′←τω+(1−τ)ω′15: **end for**

## 6. Experiments and Analysis

This section presents the evaluation of AQMDRL. We first offer the experimental setup and parameter settings, then the algorithm performance comparison, and finally, the ablation experiment.

### 6.1. Experimental Setup

In the experiments, we chose Ubuntu 18.04 for the computer operating system. The experimental simulation environment is Mininet emulation platform and RYU controller. The computer was equipped with an Intel Core i7-7700HQ CPU processor with a base frequency of 2.80 GHz, 1 TB HDD and 16 GB of RAM. The network topology was referenced to NSFNET, a backbone network formed by the National Science Foundation [39]. The network contained 14 switches and 21 links, all with a link bandwidth of 10 Mbps, as shown in Figure 2, where each switch connected to a host for receiving and sending data. The PyTorch deep learning framework implemented reinforcement learning algorithms in SDN.

### 6.2. Parameter Setting

The purpose of adjusting the parameters is for the model to obtain matching network scenarios and improve the model’s convergence performance. Consequently, our experiments determine the model’s hyper-parameters by comparing the effects of AQMDRL under various parameters. In the following experiments, to effectively reduce the link delay of the network, we set parameters $\beta_{1}$ and $\beta_{2}$ of the reward function as 10 and 1, respectively.

As a critical parameter in deep learning, the learning rate determines whether and when the objective function can converge to a local minimum. We observe the effect of the AQMDRL scheme by altering the learning rate and determine the optimal value of AQMDRL applied to the network by convergence speed and stability. In this experiment, we found that different traffic intensities have different optimal actor learning rates. We set the actor learning rate to 0.0001, 0.0003, 0.001, 0.01, and 0.1, and tested the performance of every training step. The training results are shown in Figure 3.

Figure 3 shows that the optimal actor learning rate varies with traffic intensity. Under high traffic intensities (100% and 75%), the reward curve is the best when the actor learning rate is 0.0001, as shown in Figure 3a,b. The reward curve is the best when the actor learning rate is 0.1 at low traffic intensities (50% and 25%), as shown in Figure 3c,d. In a high traffic intensity environment, because the link easily reaches the saturation state, the routing algorithm needs to adjust the link weight value more frequently. The reward is more stable after convergence at a lower actor learning rate. Based on the above results, the actor-learning rate of AQMDRL is set to 0.1, 0.1, 0.0001, and 0.0001 when the traffic intensity is 25%, 50%, 75%, and 100%, respectively. In low traffic intensity, the remaining bandwidth of the network link is sufficient, and the agent can speed up the optimal reward value exploration through a higher learning rate. However, in high traffic intensity, the agent needs a lower learning rate to explore the optimal reward value.

In addition, the parameters of step n, critic learning rate and size of replay buffer are also crucial for the experiments. Since the optimal values of these three parameters are consistent under the four traffic intensities, we choose to show our tuning results in an environment with 100% traffic intensity. The results are presented in Figure 4.

In Figure 4a, although the 8-step option reaches convergence first, the convergence value is lower, and the computational cost is higher for the 8-step compared to the 4-step option; therefore, we choose 4 as the number of steps. In Figure 4b, the reward curve is the best when the critic learning rate is 0.0003 because when the learning rate is too small or too large, the learning of AQMDRL is relatively unstable, which leads to the inability to obtain the optimal policy in a short time. In Figure 4c, the reward curve is the best when the replay buffer is 50.

The hyperparameters set in the experiment are shown in Table 1.

### 6.3. Algorithm Performance Comparison

In this section, we compare the network performance of AQMDRL with the traditional routing algorithm OSPF [2] and the DRL algorithms DDPG [25] and TD3 [30]. Evaluation metrics are convergence performance, link latency, load balancing and throughput.

First, we test the convergence performance differences of three DRL algorithms—AQMDRL, TD3, and DDPG—under different traffic intensities, as shown in Figure 5. We trained 400 episodes for each traffic intensity and tested the model’s performance in the training environment for every training step.

Reward value is used to measure the outcome of a RL action. The larger the reward value, the better the network performance is. As demonstrated in Figure 5, our AQMDRL algorithm has the fastest convergence rate and the most stable convergence performance compared to the other algorithms. DDPG has poor performance due to the problem of overestimation when blindly selecting the action with the largest Q-value. Its convergence rate is slow, and there are still large fluctuations after convergence in a high-traffic intensity scenario. TD3 minimizes the two Q-values generated by the target network, effectively solving the overestimation problem of DDPG. In the highest and lowest traffic intensity scenarios, TD3 converges faster and has a more stable convergence performance than DDPG, as shown in Figure 5a,c,d; however, it does not consider the problem of underestimation. Therefore, the curve jump is more significant under average traffic intensity, and the convergence performance is lower than that DDPG, as shown in Figure 5b. The proposed AQMDRL uses the multistep method to mitigate the overestimation and underestimation of DDPG, effectively improving the stability of the convergence in all traffic intensity scenarios. In addition, the priority caching mechanism based on SumTree improves the convergence speed of DDPG. It is worth noting that when the traffic changes more drastically, the agent must re-explore the learning operation, so the reward will be jumpy. AQMDRL will show occasional jumps but reaches convergence soon, while other algorithms take longer to restore the convergence state. The experimental results show that AQMDRL outperforms DDPG and TD3 in all traffic intensity scenarios.

Link delay and load balancing are significant for the quality of service in the network. In order to explore the performance of AQMDRL, TD3, DDPG and OSPF, we further compare the link delay and load balancing differences.

First, we focus on the link delay coefficient, that is, the sum of the reciprocal delays of each link, as shown in Equation (6), to compare the optimization of link delay in SDN networks by various algorithms. The higher the link delay coefficient, the lower the link delay, and the faster the transmission. Table 2 shows the comparison results of each episode’s average link delay coefficients at different traffic intensities.

Meanwhile, we also compare the average load-balancing coefficient of each episode at different traffic intensities. The load-balancing coefficient is the sum of the standard deviation of all link bandwidths, which is used to examine the link load-balancing degree of the network, as shown in Equation (7). The lower the load-balancing coefficient, the better the link load-balancing degree, and the higher the network link utilization. The results are shown in Table 3.

As shown in Table 2 and Table 3, the average link-delay coefficient and average load-balancing coefficient of AQMDRL are optimal in most traffic intensity scenarios. When the traffic intensity is 25%, the average link-delay coefficient of AQMDRL is slightly higher than that of TD3 and DDPG, while the average load-balancing coefficient is 15% lower than TD3 and 7% lower than DDPG. When the traffic intensity is 50%, AQMDRL has the best performance, with the average link-delay coefficient raised by 9% relative to TD3 and 6% compared to DDPG; the average load-balancing coefficient is 16% lower than TD3 and 27% lower than DDPG. When the traffic intensity is 75%, the average load-balancing coefficient of AQMDRL is worse than that of TD3 and DDPG, but the average link-delay coefficient is the highest. When the traffic intensity is 100%, the average link delay coefficient of AQMDRL is slightly lower than that of TD3, but 5% higher than that of DDPG. The average load-balancing coefficient of AQMDRL is 24% lower than TD3 and 19% lower than DDPG. Although the average load-balancing factor of OSPF is close to the three RL algorithms, the average link-delay factor is more than 10% lower than that of RL algorithms. Our results indicate that AQMDRL is superior to other reinforcement learning algorithms and OSPF. The reason is that AQMDRL improved the convergence rate and stability of the algorithm through priority experience pool mechanism and multi-step method, resulting in short trial and error time for early exploration. When there are significant changes in traffic, the agent needs to re-explore the learning operation, and AQMDRL can achieve convergence faster.

Additionally, when the traffic intensity is 25–75%, the AQMDRL’s confidence intervals of the average link-delay and load-balancing coefficient are close to TD3 and lower than DDPG. When the flow intensity is 100%, the confidence interval of AQMDRL is lower than that of TD3 and DDPG. The results show that AQMDRL collects more concentrated values and has better convergence performance than DDPG and TD3.

Next, we compare the average throughput of all hosts at different traffic intensities, as shown in Table 4. Throughput is an important indicator to measure the performance of the network. The results show that AQMDRL are optimal in all traffic intensity scenarios. In the low-traffic intensity network scenario, the network traffic does not reach the network forwarding bottleneck, and the network links are less congested. Therefore, the throughput of all algorithms has little difference, and AQMDRL is slightly higher than other algorithms. However, when the network traffic scale exceeds the network forwarding bottleneck, the throughput of AQMDRL is more than 5% higher than that of the other three algorithms due to its convergence rate and stability.

Finally, we analyze the calculation cost of the AQMDRL, TD3 and DDPG algorithms. In terms of the neural network calculation cost, both the AQMDRL and DDPG algorithms use one online network and one target network to train a mini-batch of samples, whereas TD3 requires two online networks and two target networks. Therefore, the calculation cost of TD3 is higher than that of AQMDRL and DDPG. In addition, compared with DDPG and TD3, AQMDRL increases the calculation cost of SumTree-based priority sampling O(logN), where N is the size of the replay buffer. In summary, the calculation cost of AQMDRL is higher than that of DDPG and lower than that of TD3. Meanwhile, AQMDRL has a better convergence rate and performance than DDPG and TD3.

### 6.4. Ablation Experiment

In this section, we perform ablation experiments on the proposed algorithms to verify the effectiveness of each module, mainly a multi-step module or a SumTree priority experience replay mechanism module. We compare the AQMDRL (DDPG + multi-step + SumTree) algorithm with the DDPG, SumTree-DDPG (DDPG + SumTree) and MDDPG (DDPG + multi-step) algorithms in an environment with 100% traffic intensity. The results of the ablation experiments are shown in Figure 6.

Figure 6 shows that each module of AQMDRL is crucial. SumTree-DDPG eliminates the multistep module of AQMDRL, and the convergence performance of the agent worsens. The MDDPG algorithm eliminates the SumTree-based prioritized experience playback of AQMDRL. Although it mitigates the overestimation problem of DDPG, its convergence rate is slow. The performance pf DDPG is the worst among all algorithms, with a slow convergence rate in the early stage and more jumps in the late stage. This is because DDPG blindly selects the action with the largest Q-value when selecting the action, which makes the algorithm itself have an overestimation problem. The experimental results show that our proposed AQMDRL algorithm uses a multistep method to mitigate the overestimation and underestimation of DDPG and improve the stability of agent convergence, while adding the SumTree-based priority caching mechanism to improve the convergence rate of DDPG.

## 7. Conclusions

This paper proposes AQMDRL, an automatic QoS architecture based on multistep DRL, which aims to solve the QoS problem encountered in SDN. This algorithm uses a multistep approach to solve the overestimation problem of the DDPG algorithm, which can effectively improve its stability. In addition, AQMDRL adapts a SumTree-based prioritized experience playback mechanism to optimize the convergence rate of the multistep DDPG algorithm. We design a DRL-based QoS optimization model to evaluate the SDN network quality of service and experiments to evaluate the performance of AQMDRL. The experiments show that AQMDRL significantly improves the load-balancing performance of SDN and effectively reduces the network transmission delay over existing DRL algorithms.

In future work, we will consider improving the reinforcement learning neural network to capture the characteristics of the network state better. Additionally, we will further expand the network scale and explore more methods, such as multi-agent reinforcement learning methods, multi-objective reinforcement learning methods, or multi-strategy RL methods, to improve the convergence performance of DRL algorithms in large-scale networks and optimize the network QoS more effectively.

## Figures and Tables

**Figure 1 sensors-23-00429-f001:**
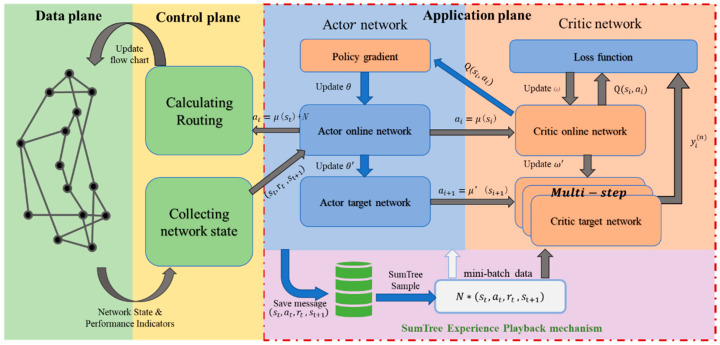
DRL-based QoS optimization architecture in SDN.

**Figure 2 sensors-23-00429-f002:**
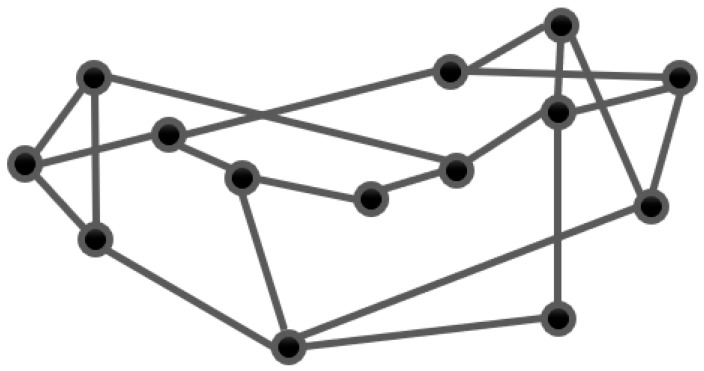
NSFNET topology. It contains 14 switches and 21 links, all with a link bandwidth of 10 Mbps, where each switch connects to a host for receiving and sending data.

**Figure 3 sensors-23-00429-f003:**
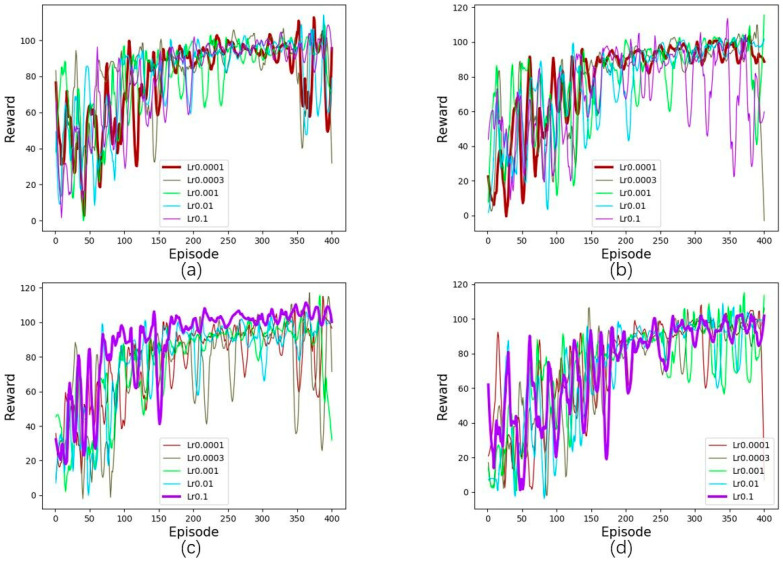
Effect of learning parameters: (**a**) the traffic intensity is 100%; (**b**) the traffic intensity is 75%; (**c**) the traffic intensity is 50%; (**d**) the traffic intensity is 25%.

**Figure 4 sensors-23-00429-f004:**
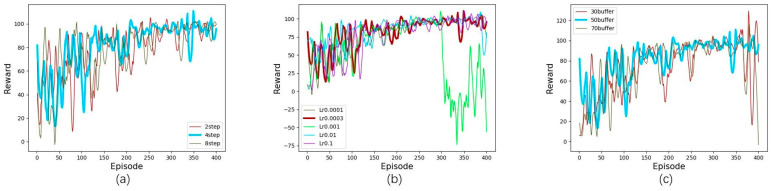
Adjustment parameter results: (**a**) number of steps; (**b**) critic learning rate; (**c**) replay buffer.

**Figure 5 sensors-23-00429-f005:**
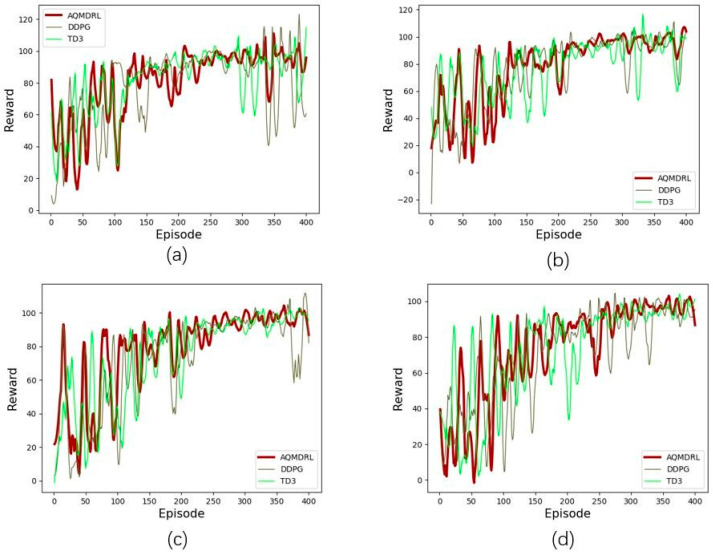
Algorithm performance comparison: (**a**) the traffic intensity is 100%; (**b**) the traffic intensity is 75%; (**c**) the traffic intensity is 50%; (**d**) the traffic intensity is 25%.

**Figure 6 sensors-23-00429-f006:**
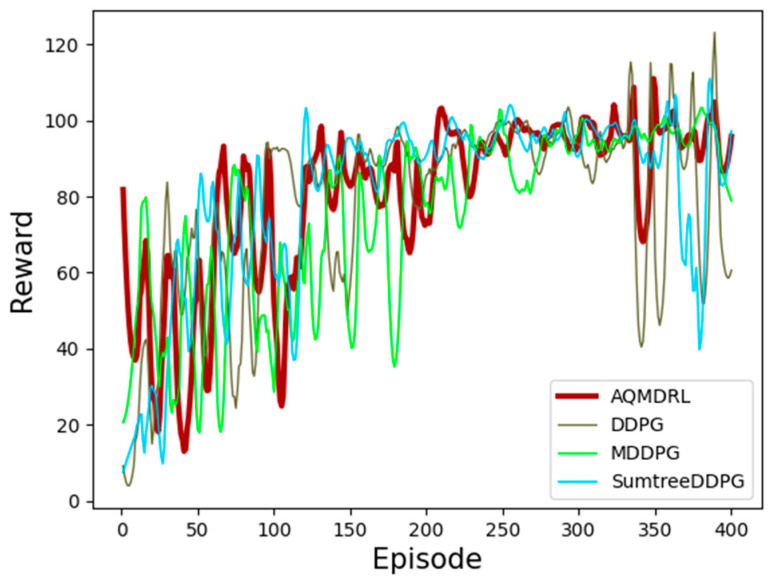
Ablation experiment results.

**Table 1 sensors-23-00429-t001:** AQMDRL training hyperparameters.

Hyperparameters	Value
Critic learning rate	0.0003
Actor learning rate (25%, 50%)	0.1
Actor learning rate (75%, 100%)	0.0001
Step n	4
Optimizer	Adam
Target update rate τ	0.005
Size of replay buffer	50
Size of mini-batch	8
Reward discount factor γ	0.99
Exploration noise	N(0, 0.1)

**Table 2 sensors-23-00429-t002:** Comparison of average link-delay coefficient in different traffic intensities.

	25% Traffic Intensity	50% Traffic Intensity	75% Traffic Intensity	100% Traffic Intensity
AQMDRL	**489 ± 40**	**531 ± 39**	**561 ± 42**	622 ± 44
TD3	478 ± 39	489 ± 38	510 ± 43	**630 ± 50**
DDPG	473 ± 43	502 ± 44	549 ± 43	591 ± 47
OSPF	352 ± 39	445 ± 48	461 ± 42	461 ± 46

**Table 3 sensors-23-00429-t003:** Comparison of average load-balancing coefficient in different traffic intensities.

	25% Traffic Intensity	50% Traffic Intensity	75% Traffic Intensity	100% Traffic Intensity
AQMDRL	**0.867 ± 0.169**	**0.647 ± 0.111**	0.802 ± 0.133	**0.697 ± 0.143**
TD3	1.025 ± 0.165	0.774 ± 0.147	0.744 ± 0.133	0.923 ± 0.298
DDPG	0.931 ± 0.303	0.884 ± 0.160	**0.612 ± 0.102**	0.858 ± 0.290
OSPF	0.934 ± 0.141	1.069 ± 0.151	0.757 ± 0.106	0.720 ± 0.129

**Table 4 sensors-23-00429-t004:** Comparison of average throughput in different traffic intensities (Mbps).

	25% Traffic Intensity	50% Traffic Intensity	75% Traffic Intensity	100% Traffic Intensity
AQMDRL	**35.65**	**68.61**	**98.02**	**116.20**
TD3	35.48	68.02	87.61	109.06
DDPG	35.48	66.76	86.89	111.84
OSPF	34.20	63.22	82.16	101.42

## Data Availability

The authors confirm that the data supporting the findings of this study are available within the article.

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
