# Peer review of "AQMDRL: Automatic Quality of Service Architecture Based on Multistep Deep Reinforcement Learning in Software-Defined Networking"

_sensors, 2022, doi:10.3390/s23010429_

Round 1

Reviewer 1 Report

1. It is necessary to modify 'slow convergence' used in line 65 to 'slow convergence rate' as in line 17.

2. In line 174, the AQMDRL was mentioned to contain the data plane and control plane of the SDN architecture, but this is not appropriate. AQMDRL can be used for SDN but does not include it.

3. Unlike b_ij^t defined in line 206, it is written in Equation 5. (ex. b_1^t)

4. The most critical point is that all performance evaluations are made for the rewards set by the author. It is necessary to show the performance improvement of the actual SDN as a result or at least prove that the reward is associated with the performance improvement of the SDN.

Reviewer 2 Report

While being used in SDN intelligent routing and scheduling mechanisms,  traditional deep reinforcement learning (DRL) algorithms present the problem of slow convergence rate and instability and so poor network quality of service (QoS) for an extended period before convergence. Aiming at the above problems, this paper proposes an automatic QoS architecture  based on multistep DRL (AQMDRL) to optimize the QoS performance of SDN. The experiments show that the AQMDRL we proposed significantly improves the convergence performance and effectively reduces the network transmission delay of SDN over existing DRL algorithms.

On the other hand, the paper should be revised by considering the following issues:

MAJOR ISSUES

+ Introduction section should be improved to give the motivation more clearly.

+ The main contributions of the paper should be clearly given as a separate subsection in the introduction section.

 + The organization of the paper should be clearly given as a separate subsection in the introduction section.

+ Especially considering the popularity of this problem, the related work should be improved. The number of references are insufficient. The related work and bibliography should be improved.

+ Most of the references in this paper are mostly recent publications (within the last 5 years) and relevant. On the other hand, the bibliography should be improved by adding most recent references instead of reference [5], [6].

+ Section “Problem Definition and System Model” should be provided clearly as a separate section.

+ Preamble information is required between section "3. AQMDRL-based QoS intelligent optimization algorithm" and subsection "3.1. System architecture".

+ Instead of abbreviation in the subsection title "3.3. DDPG", the exact name of the approach should be given.

+ In Section 3, Equation (8) should be revised because the optimization parameter is not given. By using which parameter, the optimization function can be optimized?

+Algorithm 1 should start in page 9, not the end of page 8.

+Similarly, Algorithm 2 should start in page 10, not the end of page 9.

+In Equation (23), the limits of the sum operator should be given.

+ Preamble information between section "4. Experiments and analysis" and subsection "4.1. Experimental setup" should be improved.

+ The proposed scheme performs well. The motivation behind it should be explained better.

+ The figures/schemes are generally clear. They show the data properly. It is not difficult to interpret and understand them. On the other hand, Figure 2 should be explained better by adding more information to its caption.

+ Section "4. Experiments and Analysis" should be improved. Figures should be clearly explained, especially in the text/main body of the paper.

+ What should be understood from Table 2 and Table 3?

+ The conclusion should be improved by giving the key results and main contributions more clearly.

+ Future work part should be improved in the conclusion section.

 MINOR ISSUES

+ The grammatical errors and typos should be fixed.

+ Size of Figure 5, 6 should be increased.

+ Figure 4, Table 2 and Table 3 should not exceed page margins so its size should be reduced.

+ The references in the bibliography should be given in the same style. The following link should be checked: https://www.mdpi.com/authors/references

Round 2

Reviewer 2 Report

While being used in SDN intelligent routing and scheduling mechanisms,  traditional deep reinforcement learning (DRL) algorithms present the problem of slow convergence rate and instability and so poor network quality of service (QoS) for an extended period before convergence. Aiming at the above problems, this paper proposes an automatic QoS architecture  based on multistep DRL (AQMDRL) to optimize the QoS performance of SDN. The experiments show that the AQMDRL we proposed significantly improves the convergence performance and effectively reduces the network transmission delay of SDN over existing DRL algorithms.

The authors addressed my comment on the previous version of this manuscipt considerably. On the other hand, the paper should be revised by considering the following issues:

MAJOR ISSUES

+ Introduction section should be improved to give the motivation more clearly.

+  The number of references are sufficient. Most of the references in this paper are mostly recent publications (within the last 5 years) and relevant. On the other hand, the bibliography should be improved by adding a more academic reference instead of [39] (which is a Wikipedia page).

+The authors should also consider fairness issue with QoS. For this purpose, they can consider fairness by using Jain's Fairness Index (JFI) introduced in the following papers:

- R.Jain, D-M. Chiu and W. Hawe, ”A Quantitative Measure of Fairness and Discrimination For Resource Allocation in Shared Conputer Systems,” Technical Report TR-301, DEC Research Report, Sept. 1984.

As an example of fair resource allocation with JFI, authors can consider the following paper. The proposed approach in the following paper can be considered at least as a benchmark load balancing policy for this paper.

- O. M. Gul, "Achieving Near-Optimal Fairness in Energy Harvesting Wireless Sensor Networks," 2019 IEEE Symposium on Computers and Communications (ISCC), 2019, pp. 1-6, doi: 10.1109/ISCC47284.2019.8969740.

 MINOR ISSUES

+ The grammatical errors and typos should be fixed.

+ The gap in the bottom of page 10 should be avoided by starting Section 6 in page 10 instead of page 11.

+ Table 2, Table 3 and Table 4 should not exceed page margins so its size should be reduced.

+ The references in the bibliography should be given in the same style. The following link should be checked: https://www.mdpi.com/authors/references
